# Doctor-patient communication and trust in doctors during COVID 19 times—A cross sectional study in Chennai, India

**Vijayaprasad Gopichandran**[1]*, **Kalirajan Sakthivel**[2]

**1** Department of Community Medicine, ESIC Medical College and PGIMSR, KK Nagar, Chennai, India,
**2** ESIC Medical College and PGIMSR, KK Nagar, Chennai, India

* vijay.gopichandran@gmail.com

## Abstract

### Background

The COVID 19 pandemic created a global public health crisis. Physical distancing, masks, personal protective equipment worn by the doctors created difficulties in effective doctor-patient communication.

### Objectives

This study was conducted to assess the difficulties faced by patients in communicating with their doctors due to the COVID 19 preventive measures, and its impact on the trust on their doctors.

### Methods

A cross sectional study of 359 persons attending a tertiary care center in Chennai, sampled in a non-probabilistic manner selected from the outpatient department, wards, and isolation facilities, was conducted using a questionnaire containing items covering three dimensions namely difficulties faced in accessing the health facility, difficulties in doctor-patient communication and trust in the doctors. The data were collected using Google Forms and analyzed using GNU PSPP open-source statistical software version 1.4.0.

### Results

More than 60% of the participants complained of difficulty in accessing the health facility. More than 60% had difficulties in communicating with the doctors. There was a high level of trust in doctors among more than 80% of the participants. Comparison of the mean scores revealed that accessibility was a problem across ages, sexes, education and occupation groups. Communication barriers decreased with age and increased with education, but trust increased with age, but reduced with increasing education. Multivariable linear regression analysis revealed that difficulties in communication had a negative impact on trust ($\beta = -0.63$, $p<0.001$) and increasing education had a negative impact on trust ($\beta = -0.42$, $p = 0.034$).

**Data Availability Statement:** Data are available in figshare: 10.6084/m9.figshare.14766846.

**Funding:** The authors received no specific funding for this work.

**Competing interests:** The authors declare that no competing interests exist.

## Conclusions

The COVID 19 pandemic and the preventive strategies such as lock-down, physical distancing, face mask and personal protective equipment created barriers to effective doctor patient communication and led to some compromise in trust in doctors during this time.

## Introduction

The year 2020 has endured a global health crisis in the form of the COVID 19 pandemic [1]. The disease caused by the Severe Acute Respiratory Syndrome Corona Virus 2 (SARS CoV2) spread widely across the globe and infected millions and had a case fatality rate of around 1% [2]. The pandemic entered India in late January 2020 and held fort infecting a large number of people up till late October, when the number of cases started declining [3]. In 2021, the country is facing a second wave of much larger and catastrophic proportion. Countries responded to the pandemic with closure of air travel, strict quarantine rules, lockdowns to limit spread of infection and mandatory public health measures such as wearing masks in public, temperature monitoring, hand sanitizing practices and strict isolation and treatment of the infected in dedicated Corona Virus Disease 2019 (COVID 19) care facilities. India imposed one of the harshest lockdowns in the world during the first wave in 2020 [4]. On one hand the infection was ravaging the population and on the other the stringent public health measures were having their own negative impact on people. One of the serious negative impact of the public health interventions has been restricted access to health facilities and lack of available treatments for non-COVID 19 illnesses in the public health system. Many routine public health activities suffered because of the high emphasis placed on COVID 19 prevention activities [5].

Doctors and frontline health care providers are at particularly high risk of contracting COVID 19 [6]. Therefore, there were major changes in the way front line health care workers delivered their services. Non-emergency surgeries were postponed. Frontline health workers were advised to wear masks and personal protective equipment (PPE) to safeguard themselves from the infection [7]. Physical distance was advised and so the doctor-patient encounters happened from a safe distance of about 1 meter. Doctors also limited the time they spent with the patients to effectively restrict the transmission of the illness. It is highly likely that these changes in the way that doctors delivered their services would have impacted on the effectiveness of the doctor-patient interaction.

This study was conducted to assess the difficulties faced by patients attending a tertiary care center in Chennai, in the doctor-patient communication during the peak of the COVID 19 pandemic and to study its influence on the trust in the doctor-patient relationship.

## Materials and methods

This study was conducted during July to September 2020, the peak of the COVID 19 pandemic, in Chennai, a metropolitan city in Tamil Nadu, a southern state in India. The study was conducted among persons attending a tertiary care hospital. This hospital serves employees who are covered by the Employees State Insurance Scheme, which is one of the world's largest social security schemes serving employees who earn an average monthly income of less than INR 25,000 (USD 350) [8]. The nationwide lockdown that was imposed in India on 24 March 2020 was continued in Chennai over several spells.

Sample size was estimated to establish a 50% prevalence of difficulty in doctor-patient communication with a 10% relative precision and 95% confidence level as 384 participants. Non-

probabilistic sampling, stratified by the place where the participants were interviewed, namely outpatient department, ward, COVID 19 isolation facility and hospital waiting area was performed. This was because, the patients in these locations represented various levels of severity and illness profile.

A questionnaire was developed by the study team for the purpose of this research comprising of three major domains namely, difficulties in accessing the health facility, difficulties faced in doctor-patient communication and trust in the doctors. The questionnaire responses were in a Likert format with options of 'strongly disagree', 'disagree', 'neither agree nor disagree', 'agree' and 'strongly agree'. The questionnaire items were shared with 5 experts in public health, infectious diseases and nursing and content validated. A pilot test was done among a random sample of 10 participants and based on their inputs the wordings of the questionnaires were modified to improve understanding. The questions were developed, content validated, and pilot tested in Tamil language. The final data collection was also conducted in Tamil. After analysis, the questions were translated to English for presentation.

Data collection was done using Google Forms, a web-based survey platform in the mobile hand-held device of the investigator KS. KS conducted all the interviews face to face after obtaining oral informed consent from the participants and documenting it on the Google Form. The collected data were exported to Microsoft Excel spreadsheet and cleaned by VG. The data were analyzed in the open-source statistical software GNU PSPP version 1.4.0 [9]. The characteristics of the study population and responses to the various items in the Likert scale were described as frequencies and percentages. Reliability analysis was done by calculating the Cronbach's Alpha coefficient for internal consistency of the three sub-scales namely accessibility to health facility, difficulties in doctor-patient communication and trust in doctors. Exploratory factor analysis was performed. Extraction of factors was done by principal component method; rotation was performed by Varimax method. A three-factor solution explained 67% of the variance. The Bartlett's test of sphericity showed a model fit with a statistically significant Chi square value. The KMO test also indicated sampling adequacy. The factor loadings of the exploratory factor analysis were considered as weights of the various items in the sub-scales. The crude Likert response ranging from scores of 0 to 4, were multiplied by the corresponding factor weights and a total sub-scale score was computed by adding the scores on each item in the sub-scale.

Independent sample t test and ANOVA were used to compare the mean scores on the three domains across sexes, age groups, educational and occupational groups. Multivariable linear regression analysis was performed with trust in doctors score as the dependent variable and communication difficulties, age, sex, education and occupation as independent variables.

The study was approved by the Institutional Ethics Committee of Employees State Insurance Corporation Medical College and Post Graduate Institute of Medical Sciences and Research, KK Nagar, Chennai after an expedited review process with the approval number IEC/2020/1/16 dated 29.07.2020. All interviews were conducted after obtaining oral informed consent. The Institutional Ethics Committee waived the requirement of a written informed consent in order to minimize the use of potential fomites of transmission of COVID 19 through the paper and pen on which the consent would be signed. The consent was documented in the Google Form survey platform used for data collection. Adequate privacy was ensured for each interview.

## Results

A total of 390 individuals were approached for the study out of which 360 consented to participate and responded to the questionnaire. The response rate was 92%. The 30 individuals who

**Table 1. Characteristics of the study sample.**

| S.No | Characteristic | Categories | Number | Percentage |
|------|----------------|------------|--------|------------|
| 1 | Age | < 31 yrs | 109 | 30.4% |
| | | 31–50 yrs | 175 | 48.7% |
| | | 51–60 yrs | 42 | 11.7% |
| | | >60 yrs | 32 | 8.9% |
| 2 | Sex | Male | 201 | 56% |
| | | Female | 158 | 44% |
| 3 | Education | No schooling | 87 | 24.2% |
| | | Primary School | 49 | 13.6% |
| | | Middle School | 45 | 12.5% |
| | | High School | 69 | 19.2% |
| | | Diploma | 32 | 8.9% |
| | | Under graduation | 65 | 18.1% |
| | | Postgraduation | 12 | 3.3% |
| 4 | Occupation | Unemployed | 43 | 12% |
| | | Home Maker | 81 | 22.6% |
| | | Manual Laborer | 52 | 14.5% |
| | | Skilled worker | 20 | 5.6% |
| | | Shopkeeper / Small Business | 53 | 14.8% |
| | | Clerical | 86 | 24% |
| | | Professional | 24 | 6.7% |
| 5 | Sought health care in the past 1 month | Yes | 242 | 67.4% |
| 6 | Were you diagnosed with COVID 19? | Yes | 40 | 11.1% |

did not respond gave the reasons as not willing and did not have time. Of the 360 who participated in the study, 1 questionnaire was incomplete and therefore 359 data were available and taken up for analysis. Table 1 shows the characteristics of the study sample. About half the participants (48.7%) were in the 31–50 years age group. About 30% of the participants were younger than 31 years and 20% above 50 years. More than half (56%) of the participants were men. A small proportion of 24% of the participants did not have any schooling and about 30% had studied beyond high school. About 12% were unemployed and 22% were home makers. Of the participants, 67% had sought some form of medical care in the past one month and 11% had been diagnosed with COVID 19 in the recent past.

In keeping with the main objectives of this study, the participants who consented to take part, were asked a set of 19 questions covering the key domains of accessibility, trust in doctors and problems in doctor-patient communication during the pandemic times. Their responses to the Likert scale are shown in Table 2. More than 60% of the participants responded affirmatively that they had difficulties in accessing the health facilities due to the lockdown. Similarly, more than 60% of the participants said that they faced difficulties in establishing good doctor-patient communications due to the physical distance, mask, personal protective equipment (PPE) and often did not understand the instructions given by the doctors. However, a large proportion of the participants (more than 80%) responded that they had a high level of trust in their doctors as indicated by high level of respect, trust that the doctors do what is in the patients' best interest, and the opinion that the doctor has high integrity.

The reliability of the three domains of the scale were assessed using Cronbach's alpha test of internal consistency. The Cronbach's Alpha for the accessibility dimension was 0.870, Doctor-patient communication dimension was 0.930 and trust in doctors dimension was 0.780. Therefore, all the three dimensions had acceptable levels of internal consistency reliability. The

**Table 2. Responses to questions related to health care access, doctor-patient communication and trust in doctors during COVID 19 times.**

| S. No | Question | Disagree | Somewhat Disagree | Neither agree nor disagree | Somewhat agree | Agree |
|---|---|---|---|---|---|---|
| 1 | As all nearby clinics were closed due to lockdown it was difficult to access health care | 45 (12.5%) | 4 (1.1%) | 74 (20.6%) | 21 (5.8%) | 215 (59.9%) |
| 2 | As all transport facilities were suspected it was difficult to access health facilities | 45 (12.5%) | 1 (0.3%) | 73 (20.3%) | 23 (6.4%) | 217 (60.4%) |
| 3 | As doctors practice physical distancing, it was difficult interacting with them | 89 (24.8%) | 2 (0.6%) | 23 (6.4%) | 105 (29.2%) | 140 (39%) |
| 4 | As doctors wear mask and PPE it is difficult to interact with them | 89 (24.8%) | 2 (0.6%) | 22 (6.1%) | 105 (29.2%) | 141 (39.3%) |
| 5 | Doctors do not spend much time with patients due to fear of infection | 191 (53.2%) | 14 (3.9%) | 19 (5.3%) | 55 (15.3%) | 80 (22.3%) |
| 6 | Doctors do not touch the patients and so treatment feels inadequate | 148 (41.2%) | 8 (2.2%) | 35 (9.7%) | 75 (20.9%) | 93 (25.9%) |
| 7 | Due to the physical distance and the PPE we are unable to understand the instructions of the doctors | 96 (26.7%) | 5 (1.4%) | 23 (6.4%) | 113 (31.5%) | 122 (34%) |
| 8 | Due to too much focus on COVID 19 doctors are not paying much attention to other illnesses | 234 (65.2%) | 11 (3.1%) | 28 (7.8%) | 22 (6.1%) | 64 (17.8%) |
| 9 | As doctors have reduced giving injections, treatment feels inadequate | 83 (23.1%) | 4 (1.1%) | 182 (50.7%) | 25 (7%) | 65 (18.1%) |
| 10 | Nowadays we do not have a choice of doctors or hospitals | 48 (13.4%) | 2 (0.6%) | 60 (16.7%) | 56 (15.6%) | 193 (53.8%) |
| 11 | Nowadays we are unable to trust that everything will be alright if we consult the doctor | 246 (68.5%) | 8 (2.2%) | 10 (2.8%) | 51 (14.2%) | 44 (12.3%) |
| 12 | I trust that the doctor has my best interest in mind | 49 (13.6%) | 12 (3.3%) | 7 (1.9%) | 43 (12%) | 248 (69.1%) |
| 13 | I trust that the doctor is honest | 24 (6.7%) | 7 (1.9%) | 11 (3.1%) | 19 (5.3%) | 298 (83%) |
| 14 | I trust that the doctor's advice is for my benefit | 29 (8.1%) | 4 (1.1%) | 12 (3.3%) | 21 (5.8%) | 293 (81.6%) |
| 15 | I trust that the doctor works for my best interest even during the pandemic times | 34 (9.5%) | 4 (1.1%) | 22 (6.1%) | 33 (9.2%) | 266 (74.1%) |
| 16 | As these are pandemic times I can understand why doctors and hospitals are acting in a precautionary manner | 40 (11.1%) | 4 (1.1%) | 18 (5%) | 45 (12.5%) | 252 (70.2%) |
| 17 | As doctors and hospitals are also suffering a financial crisis, I understand the high cost of treatment | 278 (77.4%) | 7 (1.9%) | 34 (9.5%) | 27 (7.5%) | 13 (3.6%) |
| 18 | As doctors are overworked, I can understand if they are rude to me. | 111 (30.9%) | 1 (0.3%) | 2 (0.6%) | 64 (17.8%) | 181 (50.4%) |
| 19 | I respect the doctor a lot | 22 (6.1%) | 3 (0.8%) | 7 (1.9%) | 35 (9.7%) | 292 (81.3%) |

findings of the exploratory factor analysis are shown in Table 3. It is seen that the three dimensions are separated appropriately with good factor loadings all above 0.4, indicating good structural validity of the scale. The respective factor loadings were considered as the weight for each of the items and the Likert response from 0–4 was multiplied by the factor weight of that item and then added up to generate the total score in that dimension for each participant.

Table 4 shows the mean score in each dimension. It is seen that the mean score was high in both the inaccessibility domain and the trust in the doctors domain, whereas it was around the middle in the doctor-patient communication difficulties domain.

In order to study the various factors influencing the score in each domain, the mean scores were compared between sexes, age groups, education groups and type of occupation. This is shown in Table 5. It is seen that women had greater trust in physicians than men, whereas there was no significant sex difference in the accessibility and communication barriers. With increasing age there was increasing trust in the doctor, reducing difficulties in doctor-patient

**Table 3. Exploratory factor analysis showing the grouping of variables into three dimensions and their factor weights.**

| Items | Trust in the doctor | Accessibility | Doctor-patient communication |
|---|---|---|---|
| As all nearby clinics were closed due to lockdown it was difficult to access health care | | .94 | |
| As all transport facilities were suspected it was difficult to access health facilities | | .92 | |
| As doctors practice physical distancing, it was difficult interacting with them | | | .93 |
| As doctors wear mask and PPE it is difficult to interact with them | | | .94 |
| Doctors do not spend much time with patients due to fear of infection | | | .50 |
| Doctors do not touch the patients and so treatment feels inadequate | | | .71 |
| Due to the physical distance and the PPE we are unable to understand the instructions of the doctors | | | .92 |
| Due to too much focus on COVID 19 doctors are not paying much attention to other illnesses | | | .39 |
| As doctors have reduced giving injections, treatment feels inadequate | | | .67 |
| Nowadays we do not have a choice of doctors or hospitals | | .75 | |
| Nowadays we are unable to trust that everything will be alright if we consult the doctor | -.75 | | |
| I trust that the doctor has my best interest in mind | .77 | | |
| I trust that the doctor is honest | .92 | | |
| I trust that the doctors advice is for my benefit | .93 | | |
| I trust that the doctor works for my best interest even during the pandemic times | .91 | | |
| As these are pandemic times I can understand why doctors and hospitals are acting in a precautionary manner | .87 | | |
| As doctors are overworked, I can understand if they are rude to me. | .65 | | |
| I respect the doctor a lot | .86 | | |

communication and increasing inaccessibility to health facilities, all of which were statistically significant. With increasing education levels, trust in the doctors seemed to reduce, difficulties in doctor-patient communication seemed to increase and inaccessibility to health facilities decreased, and all these were statistically significant associations. Such a strong and clear association was not seen with occupation.

Fig 1 shows the association between problems in doctor-patient communication and the trust in the doctors. The scatter plot shows a negative correlation of reducing trust in the doctor with increasing barriers in doctor-patient communication.

Multivariable linear regression to study the association between difficulty in doctor-patient communication and trust in physicians after adjusting for age, education and occupation confirmed the negative association between difficulty in doctor-patient communication and trust in the physicians. It was further seen that age and occupation did not have an influence on trust, but education was also negatively associated with trust, increasing education leading to lesser trust in the doctors. This multivariate linear regression is shown in Table 6.

## Discussion

This cross-sectional survey among patients attending a tertiary care facility in Chennai showed that a majority of them faced difficulties in accessing health care facilities due to the lockdown.

**Table 4. Weighted scores on the dimensions of accessibility, communication and trust.**

| S.No | Dimension (minimum and maximum possible scores) | Mean Score | SD |
|---|---|---|---|
| 1 | Inaccessibility to Health Facilities (0–10.44) | 7.81 | 3.89 |
| 2 | Doctor-Patient Communication problems (0–20.24) | 10.88 | 6.87 |
| 3 | Trust in the doctor (0–20.64) | 18.96 | 7.52 |

**Table 5. Comparison of accessibility, doctor-patient communication and trust in doctors based on characteristics of the participants.**

| S. No | Characteristic | Categories | Trust in Doctor Scores (mean ± SD) | p value | Doctor patient communication problems (mean ± SD) | p value | Accessibility (mean ± SD) | p value |
|---|---|---|---|---|---|---|---|---|
| 1 | Sex | Male | 18.28 ± 8.20 | <0.001* | 11.09 ± 7.14 | 0.110 | 7.88 ± 3.44 | 0.340 |
| | | Female | 19.82 ± 6.46 | | 10.61 ± 6.52 | | 7.72 ± 3.10 | |
| 2 | Age | < = 30 yrs | 17.77 ± 7.78 | 0.020* | 12.17 ± 6.82 | 0.001* | 7.53 ± 3.57 | 0.360 |
| | | 31–50 yrs | 18.87 ± 7.94 | | 10.84 ± 6.64 | | 7.78 ± 3.49 | |
| | | 51–60 yrs | 20.28 ± 5.21 | | 10.60 ± 6.85 | | 8.11 ± 3.17 | |
| | | >60 yrs | 22.12 ± 4.79 | | 6.77 ± 6.85 | | 8.65 ± 2.69 | |
| 3 | Education | Uneducated | 21.16 ± 5.78 | < 0.001* | 8.87 ± 6.70 | < 0.001* | 8.46 ± 2.85 | 0.047* |
| | | Primary School | 21.06 ± 6.13 | | 9.65 ± 6.98 | | 8.34 ± 3 | |
| | | Middle School | 18.79 ± 8.99 | | 10.14 ± 7.22 | | 8.04 ± 3.35 | |
| | | High School | 19.30 ± 7.95 | | 11.26 ± 6.81 | | 7.41 ± 3.77 | |
| | | Diploma | 17.92 ± 6.81 | | 11.55 ± 6.86 | | 7.66 ± 3.03 | |
| | | Undergraduation | 15.42 ± 8.10 | | 13.52 ± 6.20 | | 7.27 ± 3.26 | |
| | | Post Graduation | 14.93 ± 5.81 | | 15.04 ± 4.10 | | 5.70 ± 4.14 | |
| 4 | Occupation | Unemployed | 18.05 ± 7.81 | <0.001* | 11.34 ± 6.92 | 0.093 | 7.80 ± 3.27 | 0.264 |
| | | Home Maker | 21.34 ± 5.28 | | 9.51 ± 6.87 | | 8.18 ± 3.02 | |
| | | Manual Labourer | 18.71 ± 8.84 | | 10.56 ± 7.04 | | 8.17 ± 3.17 | |
| | | Skilled Worker | 21.03 ± 6.02 | | 9.95 ± 6.23 | | 7.20 ±. 4.18 | |
| | | Shopkeeper / Small Business | 20.04 ± 6.57 | | 10.18 ± 6.26 | | 7.15 ± 3.13 | |
| | | Clerical | 17.23 ± 8.38 | | 11.99 ± 7.31 | | 8.10 ± 3.42 | |
| | | Professional | 15.10 ± 7.50 | | 13.73 ± 5.71 | | 6.74 ± 3.42 | |

*statistically significant $p<0.05$, independent sample t test and ANOVA performed for comparison of means

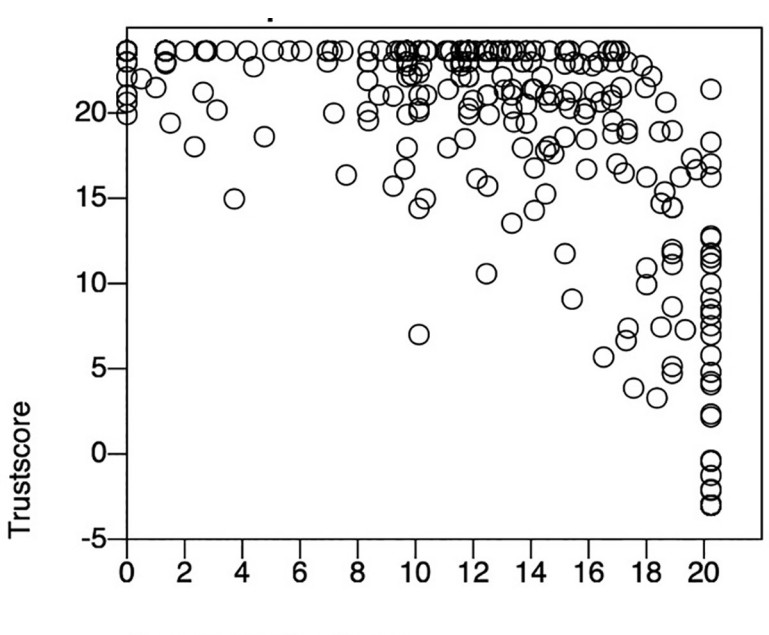

**Fig 1. Association between problem with communication and trust in doctors.**

**Table 6. Association between doctor patient communication and trust in the doctors.**

| Factors influencing trust scores | Beta Coefficient | 95% CI | p value |
|---|---|---|---|
| Doctor-patient communication score | -0.630 | -0.730 to -0.540 | <0.001* |
| Age | 0 | -0.05 to 0.05 | 0.932 |
| Sex | 0.840 | -0.480 to 2.170 | 0.213 |
| Education | -0.420 | -0.810 to -0.030 | 0.034* |
| Occupation | -0.07 | -0.440 to 0.300 | 0.695 |

*statistically significant p<0.05, multiple linear regression analysis

Many of them found it difficult to communicate with their doctors due to the physical distancing, personal protective equipment and limited time spent with them due to COVID 19 advisories. Despite this inaccessibility and difficulty in communicating with the doctors, their trust in doctors remained high even during the COVID 19 pandemic times. Further it was noted that women had greater trust in the doctors. With increasing age, trust in doctors increased but difficulty in communication decreased and with increasing education levels trust in doctors decreased and difficulties in communication increased. There was a relatively strong negative correlation between doctor-patient communication barriers and trust in the doctors even after adjusting for age, sex, education and occupation.

COVID 19 laid bare the weakness of the public health system in India. The lockdown impaired the access to healthcare facilities that were already inaccessible to many poor and marginalized people in the country. Many parts of the country faced serious limitations in access to health care during the pandemic for non-COVID 19 illnesses [10]. There were even reports of interruption of treatment for chronic non communicable diseases due to access issues [11]. Though this was a universal phenomenon, the urban slums in low- and middle-income countries were worse affected by this lack of access to health facilities [12]. Chennai city was a hot spot of transmission of COVID 19, and lockdowns were imposed very early during the pandemic. This lack of access related to the lockdown was reported in this study too. It was observed that this lack of access was perceived by people of both sexes, all age groups and across all educational and occupational classes. Even people who had their own private vehicles, found it difficult to get past the strict curfew and make it to health facilities.

Several studies have reported the difficulty in doctor-patient communication during the COVID 19 times. A study from Africa pointed out that patients perceived that physical distancing and personal protective equipment impaired the doctor-patient relationship [13]. Patients, especially the elderly, felt apprehensive communicating with doctors covered in PPE and this worsened their anxiety in the hospital [14]. Firstly, the mask and PPE covered the human face of the doctor. This created a sense of disconnect between the doctor and the patient. Covering the face with the mask prevented the doctors from expressing any facial cues including empathy, compassion, kindness all of which could be very effectively communicated by facial expressions. Moreover, individuals who are hearing and speech disabled, depend largely on lip reading for communicating with their doctor. The mask and head gear prevented these patients from reading the lips of their doctors. These greatly impaired the doctor-patient communication [15]. In this study also patients reported that the mask, PPE and physical distancing impaired effective communication with their doctors. It would be natural to expect that these communication issues would worsen with increasing age as older individuals are more likely to have vision and hearing difficulties. However, it was observed in this study that the communication problems were reported more among the younger individuals and it reduced with increasing age. One possible explanation for this could be that the younger

individuals were more demanding and expecting of clear communication from their doctors compared to the elderly. It is also possible that the lack of clear communication was routine among the elderly, and they did not find it different with the mask, PPE and physical distance. One other explanation could be that the elderly experienced a sense of gratitude for having received any kind of medical attention despite the lockdown and this made them ignore the difficulties in communication. In medical institutions there are strong lines of hierarchy with the doctor on the top and other allied health professionals below them. However, the wearing of the mask and PPE homogenized all staff as it became difficult to identify the person inside the mask and PPE. This is probably one of the reasons why communication barriers were not perceived by the elderly as much as the younger persons.

The third important finding of the study was high levels of trust in the doctors, despite poor accessibility and difficulty in doctor-patient communication. One other previous empirical evaluation of trust in doctors in Tamil Nadu, close to the study setting, also revealed a high level of trust in doctors [16]. While there have been reports of eroding trust in physicians and the health system in the United States during the COVID 19 times because of a lack of consistent public health messaging on hydroxychloroquine and masks in the country, such a pattern of lack of trust has not been seen in India [17, 18]. The dimensions of trust in physicians in a low- and middle-income country setting like India have been explored in the past and the key dimensions are perceived competence, assurance of treatment, respect and loyalty [19]. It is seen that even though many patients were deprived of the assurance of good quality treatment, the overriding dimensions of respect and loyalty, ensured that they retained the basic trust in doctors. In this study the items including, 'I trust that the doctor is honest', 'I trust that the doctor works for my best interest even during the pandemic times' and 'I respect the doctor a lot' had a high rate of affirmative response. This indicated the high level of trust in the doctors. It was also observed in this study that women had greater trust than men, trust in doctors increased with age, and people with higher education had lower trust levels. Those who were home makers, unemployed and manual laborers had greater trust compared to those who were in business, clerical work and professional jobs. It is worth noting that the influence of age, sex and occupation on trust in physicians was nullified in the multivariable model, leaving only education and barriers in communication as factors influencing trust.

In a previous study of factors affecting trust in the doctor-patient relationship, it was noted that the doctor-patient communication including a personal involvement of the doctor with the patient greatly influenced the trust [20]. Based on this premise, this study attempted to explore the association between doctor-patient communication during COVID 19 times and the trust in doctors. A relatively strong inverse association was established in this study. Those who perceived greater difficulties in communication with their physician also reported lesser trust in their physicians. Even after adjusting for age, sex, education and occupation, it was seen that difficulty in communication remained negatively associated with trust in the doctors.

The strength of this study is that it was conducted during the peak of the COVID 19 pandemic among patients attending a tertiary care center to understand a crucial aspect of the doctor-patient relationship during the difficult pandemic times. The calculated sample size was 384, however, only a sample size of 359 could be achieved and analyzed. Sampling was stratified for the location in the hospital where the participants were interviewed. However, this data was not captured in the questionnaire due to a practical error and hence the impact of the location of data collection (non-COVID versus COVID) on trust and communication barriers could not be analyzed. The study also could not provide information on whether the trust and communication barriers depended on the severity of the disease. Another possible limitation could be a socially desirable response bias, as the interviews were conducted by the researchers in a health care facility. Despite these limitations, the study helps document an

important dimension of the doctor patient relationship during the COVID 19 pandemic, namely communication and trust. Future studies should explore the dimensions of doctor-patient relationships during the pandemic times using qualitative methods, which are more suited for in-depth understanding of such experiences.

The COVID 19 experience has taught us that during pandemic times, while it is important to focus on public health measures, it is equally important to keep people at the center of the health care enterprise. All public health and disease prevention interventions must be people centered and focus on the welfare of the people [21]. This study further contributed to this idea by clearly indicating that doctor-patient communication and trust are very important considerations during pandemic times.

## Acknowledgments

The authors would like to acknowledge the participants in the study for their valuable insights in their experiences of doctor-patient relationships during the COVID 19 pandemic.

## Author Contributions

**Conceptualization:** Vijayaprasad Gopichandran, Kalirajan Sakthivel.

**Data curation:** Kalirajan Sakthivel.

**Formal analysis:** Vijayaprasad Gopichandran.

**Methodology:** Vijayaprasad Gopichandran, Kalirajan Sakthivel.

**Software:** Vijayaprasad Gopichandran, Kalirajan Sakthivel.

**Supervision:** Vijayaprasad Gopichandran.

**Validation:** Vijayaprasad Gopichandran, Kalirajan Sakthivel.

**Writing – original draft:** Vijayaprasad Gopichandran.

**Writing – review & editing:** Vijayaprasad Gopichandran, Kalirajan Sakthivel.

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
