## [Decision Letter · Decision Letter 0]

25 Apr 2021

PONE-D-21-08356

Doctor-patient communication and trust in doctors during COVID 19 times – a cross sectional study in Chennai, India

PLOS ONE

Dear Dr. Gopichandran,

Thank you for submitting your manuscript to PLOS ONE. After careful consideration, we feel that it has merit but does not fully meet PLOS ONE’s publication criteria as it currently stands. Therefore, we invite you to submit a revised version of the manuscript that addresses the points raised during the review process.

We look forward to receiving your revised manuscript.

Kind regards,

Prof. Ritesh G. Menezes, M.B.B.S., M.D., Diplomate N.B.

Academic Editor

PLOS ONE

Journal Requirements:

Please include additional information regarding the survey or questionnaire used in the study and ensure that you have provided sufficient details that others could replicate the analyses. For instance, if you developed a questionnaire as part of this study and it is not under a copyright more restrictive than CC-BY, please include a copy, in both the original language and English, as Supporting Information.

We note that you have indicated that data from this study are available upon request. PLOS only allows data to be available upon request if there are legal or ethical restrictions on sharing data publicly. For information on unacceptable data access restrictions, please see http://journals.plos.org/plosone/s/data-availability#loc-unacceptable-data-access-restrictions.

3a) If there are ethical or legal restrictions on sharing a de-identified data set, please explain them in detail (e.g., data contain potentially identifying or sensitive patient information) and who has imposed them (e.g., an ethics committee). Please also provide contact information for a data access committee, ethics committee, or other institutional body to which data requests may be sent.

3b) If there are no restrictions, please upload the minimal anonymized data set necessary to replicate your study findings as either Supporting Information files or to a stable, public repository and provide us with the relevant URLs, DOIs, or accession numbers. Please see http://www.bmj.com/content/340/bmj.c181.long for guidelines on how to de-identify and prepare clinical data for publication. For a list of acceptable repositories, please see http://journals.plos.org/plosone/s/data-availability#loc-recommended-repositories.

Reviewers' comments:

Reviewer's Responses to Questions

**Comments to the Author**

1. Is the manuscript technically sound, and do the data support the conclusions?

Reviewer #1: Yes

Reviewer #2: Yes

Reviewer #3: No

Reviewer #4: Yes

Reviewer #5: Yes

Reviewer #6: Yes

Reviewer #7: Yes

Reviewer #8: Yes

Reviewer #9: Yes

2. Has the statistical analysis been performed appropriately and rigorously? 

Reviewer #1: Yes

Reviewer #2: Yes

Reviewer #3: Yes

Reviewer #4: Yes

Reviewer #5: Yes

Reviewer #6: I Don't Know

Reviewer #7: Yes

Reviewer #8: Yes

Reviewer #9: I Don't Know

3. Have the authors made all data underlying the findings in their manuscript fully available?

Reviewer #1: Yes

Reviewer #2: Yes

Reviewer #3: Yes

Reviewer #4: Yes

Reviewer #5: Yes

Reviewer #6: Yes

Reviewer #7: Yes

Reviewer #8: Yes

Reviewer #9: Yes

4. Is the manuscript presented in an intelligible fashion and written in standard English?

Reviewer #1: Yes

Reviewer #2: Yes

Reviewer #3: Yes

Reviewer #4: Yes

Reviewer #5: Yes

Reviewer #6: Yes

Reviewer #7: No

Reviewer #8: Yes

Reviewer #9: Yes

5. Review Comments to the Author

Reviewer #1: Good study. However the arrangements of sentences and paragraphs are not in sequential manner. The title and the content are not matching. As per the content more data regarding the doctor patient communication must have been analyzed.

Reviewer #2: The article investigated difficulties faced by patients in communicating with their doctors due to the COVID 19 preventive measures. This is an important issue. Although some issues found which need to be addressed prior to its publication.

1. Avoid using long sentences. This have caused grammatical errors and readability of the article.

2. Kindly use full form of abbreviations on first use.

3. Use MeSH terms as keywords.

4.In introduction every paragraph must have a theme. Paragraph 2 and 4 is very short. Club them with preceding paragraph. Arrange introduction as following: background characteristics, knowledge gaps exists, how your study is going to address those gaps. Cite important references used in discussion in introduction.

5. Avoid using terms like heart of the city. If your hospital were in heart were was the liver, kidneys and lungs????

6. If used a non-probabilistic sampling then why design effect was not used for sample size calculation.

7. Tables should be self-explanatory. Kindly write abbreviations used in the tables in the foot notes. Statistical analysis used in each table should either be used in title or footnotes. Indicate 'N' of each table.

8. If several prior studies were available on the issue. Then why they were not used in sample size calculation.

9. The article did not follow uniform referencing style. Kindly manually redo it as per the journals guidelines.

Reviewer #3: Doctor patient communication and trust is a complex process where social and emotional factors, empathy and attentive listening are important parameters, qualitative techniques may provide more information. I think it covers some points related to quality of care. Only a part of interaction was assessed without any comparison group. sampling technique is inappropriate and not representing the categories of morbidities.

Reviewer #4: Reviewer’s comment for the article entitled “Doctor-patient communication and trust in doctors during COVID 19 times – a cross sectional study in Chennai, India.”

The research question is quite interesting, innovative and satisfying the current need where continuous efforts are explored for doctor-patient communication and trust in pandemic situation. The article is well written and addressing major aspects in focus. The rationale for the proposed study seems clear and valid. The study has a clear methodology and interesting findings which would be positively utilized by the readers at different forums. Results can be understood simply and categorized properly. It includes relevant information. The data and analyses support the claims and findings. Discussion portion is clearly written. The authors are free to receive a suggestion to include recommendations to overcome limitations of the study which would be useful for future explorations by reader community. The study shows sufficient potential and up to the standards of the journal.

Reviewer #5: The paper is well written and covers a very relevant and important area during this pandemic. The analysis has been done reasonably well and results presented in an intelligible manner. The results are very pertinent to changing policies and behaviour.

Reviewer #6: This is a timely manuscript assessing the difficulties faced by patients in communicating with their doctors due to the COVID 19 preventive measures, and its impact on the trust on their doctors. The study was conducted during the peak of the pandemic in a large Indian metro city in a tertiary care hospital that caters to a specific group of patients. The participants were identified and recruited in four settings (in patients, OPD patients, Covid isolation facilities and the hospital waiting areas) and authors rightly explain the importance of settings where they were recruited in terms of severity and urgency of their symptoms.

The most intriguing finding of the study is that with increasing age there was increasing trust in the doctor but less difficulties in doctor-patient communication. One would expect otherwise especially the communication barrier increasing with increasing age of the patient. The authors elaborate on three possible reasons for this finding which are interesting and plausible in the context of Indian healthcare system. However, the fact that the communication with elderly patients might always have been challenging and therefore the elderly did not experience any added barriers in communicating with the doctors raises important questions regarding doctor patient communication during non-covid times. To what extent this finding could be also influenced by the fact that the elderly in particular felt grateful to be seen by someone in a health facility in these trying times when they might have tried hard to reach the hospital for several days? Could this sense of gratitude combined with high degree of respect towards doctors in general in Indian society at least in the minds of older patients influenced their perception of reduced communication barrier? This aspect can be explored further in the discussion section.

In follow-up of the point above, healthcare providers in PPE and masks during Covid 19 pandemic probably made it impossible for the patients to assess whether the particular HCW interacting with them is a doctor, a medical student and intern or a nurse. There is hierarchy among different health care professionals and patients also respect and trust these different groups of health care professionals differently. PPE and overalls in some ways homogenized all HCWs which can otherwise be easily stratified by patients based on the age, socio-economic status based on jewelry and clothing wore under the apron etc. I wonder what impact this phenomenon (of homogenization of HCWs in PPEs) could have had on patients’ level of trust in doctors. Could authors reflect on this aspect further and particularly its implications for doctor patient relationship in non-pandemic settings.

Women patients trusted doctors more than men. How can this be explained especially if it is not linked with education? Is it because women in general are less often in position of power or less likely to challenge the individuals with authority like doctors and therefore tend to put higher respect and trust in doctors?

I wonder whether the authors analyzed the responses in three domains in relation to the settings in which the participants were recruited in this study. How was the experience of those seen in outpatient department different than those who were admitted or were in isolation facility or were just in waiting areas? My suspicion is that the context in which they answered the questions combined with their symptoms and severity of those symptoms might pose different challenges in doctor patient communication and might also affect the level of trust they put in their healthcare providers. I would also expect difference in challenges encountered to access health services for participants in these four distinct settings, not just in terms of reaching the healthcare facility but also once they entered the facility while they were being triaged or diverted in different streams. Could authors elaborate on why they did not pursue this line of analysis?

Reviewer #7: Dear Authors,

The submitted manuscript, with its objectives is currently of utmost importance in the COVID era. This may help in formulating policies for future pandemics.

However, the following points need to be rectified in the manuscript

1. There are two objectives- first is to formulate a questionnaire using factor analysis and then execute the same. Hence, this needs to be clarified and subdivided in the paper

2. The flow of paper needs to be streamlined. The factor analysis to be explained first, then the execution of the study using this questionnaire.

3. Language and fluidity needs to be looked into.

4. Other comments have been made as track changes in the attached PDF.

Regards,

KS

Reviewer #8: The title of the study is more contemporary addressing the current issues that has struck the globe. There is an attempt with success to explore the outcomes from the COVID imposed behaviour which is also made mandatory by the Government and Statutes as 'Covid appropriate behaviour'.It is further appreciated that in a Metropolitan City of Southern India, the Covid appropriate behaviour from the part of the doctors and medical professionals had reduced the effective doctor-patient relationship. The authors could have explored the probable measures to overcome these obstacles, especially in a time when the globe is further being made to face the second wave ofthe pandemic.

Reviewer #9: The authors have used Cronbach’s Alpha coefficient for internal consistency of parameters. This is not a statistical method but I do not know details of how this method works. I also do not know varimax rotational method although the meaning of factor analysis to study relationship of factors is understood. Application of Bartlett's test of sphericity to test whether the authors' model is fit or not is also not understood by me.

I have attached my review comments. On the whole the paper appears to be fine and requires only minor corrections.

6. PLOS authors have the option to publish the peer review history of their article (what does this mean?). If published, this will include your full peer review and any attached files.

Reviewer #1: No

Reviewer #2: **Yes: **Bijit Biswas

Reviewer #3: No

Reviewer #4: No

Reviewer #5: **Yes: **Aneesh Basheer

Reviewer #6: No

Reviewer #7: No

Reviewer #8: **Yes: **Dr. Arun M

Reviewer #9: No

---

## [Author Response · Author response to Decision Letter 0]

28 Apr 2021

Reviewer #1: Good study. However the arrangements of sentences and paragraphs are not in sequential manner. The title and the content are not matching. As per the content more data regarding the doctor patient communication must have been analyzed.

Response: Thank you for your warm comment about our study. We have now read the manuscript closely again and have re-arranged some of the sentences and paragraphs to improve sequencing. The title is “Doctor-patient communication and trust in doctors during COVID 19 times – a cross sectional study in Chennai, India”. The content of the manuscript describes the various challenges that patients faced in doctor-patient communication and also the level of trust they had in their doctors. Therefore the title and the content are matching. 

Reviewer #2: The article investigated difficulties faced by patients in communicating with their doctors due to the COVID 19 preventive measures. This is an important issue. Although some issues found which need to be addressed prior to its publication.

1. Avoid using long sentences. This have caused grammatical errors and readability of the article.

Response: Thank you for this useful comment. We have substantially edited several long sentences. 

2. Kindly use full form of abbreviations on first use.

Response: We have now expanded all abbreviations on first use. 

3. Use MeSH terms as keywords.

Response: We have now changed the keywords to MESH terms. 

4.In introduction every paragraph must have a theme. Paragraph 2 and 4 is very short. Club them with preceding paragraph. Arrange introduction as following: background characteristics, knowledge gaps exists, how your study is going to address those gaps. Cite important references used in discussion in introduction.

Response: Thank you for this very useful comment. We have now reorganized our paragraphs as per the suggestion provided. We have also arranged them according to the following themes – Introduction to COVID 19 in India, Impact of COVID 19 on health care in India, especially impact on doctor-patient communication, Objective of the study. 

5. Avoid using terms like heart of the city. If your hospital were in heart were was the liver, kidneys and lungs????

Response: We have removed the use of the term ‘heart of the city’. 

6. If used a non-probabilistic sampling then why design effect was not used for sample size calculation.

Response: We believe that the difficulty in communication as well as the compromise in trust in the doctors must have been a universal phenomenon with very little variability, as the difficulties were faced by everyone. Since we expected a substantial homogeneity in response, we did not see the need to have a design effect, as the lack of random sampling is less likely to have altered the findings due to the homogeneity. 

7. Tables should be self-explanatory. Kindly write abbreviations used in the tables in the foot notes. Statistical analysis used in each table should either be used in title or footnotes. Indicate 'N' of each table.

Response: We have now mentioned abbreviations used for each table. We have mentioned the statistical analysis used in the footnotes. 

8. If several prior studies were available on the issue. Then why they were not used in sample size calculation.

Response: Prior studies have been largely qualitative and have looked at communication barriers. There have not been any quantitative estimates of trust and communication difficulties among patients. Therefore these could not be used in calculation of sample size.

9. The article did not follow uniform referencing style. Kindly manually redo it as per the journals guidelines.

Response: We have redone the references manually as per the journal style. 

Reviewer #3: Doctor patient communication and trust is a complex process where social and emotional factors, empathy and attentive listening are important parameters, qualitative techniques may provide more information. I think it covers some points related to quality of care. Only a part of interaction was assessed without any comparison group. sampling technique is inappropriate and not representing the categories of morbidities.

Response: We agree that qualitative methods would have been more useful to capture the experiences of the patients in greater depth. Given the pandemic situation, and a fear among people to sit and have lengthy discussions with others, we decided to adopt a survey method, which involved significantly lesser time. 

There is an internal comparison group. These various internal comparisons are highlighted in Table 5. 

The sampling technique stratified the patients based on which place they attended, the general clinic, the COVID 19 clinic of the COVID 19 isolation ward. We believe this represents people who are most likely to have communication issues with their physicians. We have not stratified based on morbidity, as we believe that across different levels of morbidity there existed similar communication problems. 

Reviewer #4: Reviewer’s comment for the article entitled “Doctor-patient communication and trust in doctors during COVID 19 times – a cross sectional study in Chennai, India.”

The research question is quite interesting, innovative and satisfying the current need where continuous efforts are explored for doctor-patient communication and trust in pandemic situation. The article is well written and addressing major aspects in focus. The rationale for the proposed study seems clear and valid. The study has a clear methodology and interesting findings which would be positively utilized by the readers at different forums. Results can be understood simply and categorized properly. It includes relevant information. The data and analyses support the claims and findings. Discussion portion is clearly written. The authors are free to receive a suggestion to include recommendations to overcome limitations of the study which would be useful for future explorations by reader community. The study shows sufficient potential and up to the standards of the journal.

Response: Thank you for your positive review of our manuscript. We have included a recommendation section for improving the limitations of this study. 

Reviewer #5: The paper is well written and covers a very relevant and important area during this pandemic. The analysis has been done reasonably well and results presented in an intelligible manner. The results are very pertinent to changing policies and behaviour.

Response: Thank you for your positive review of the manuscript. 

Reviewer #6: This is a timely manuscript assessing the difficulties faced by patients in communicating with their doctors due to the COVID 19 preventive measures, and its impact on the trust on their doctors. The study was conducted during the peak of the pandemic in a large Indian metro city in a tertiary care hospital that caters to a specific group of patients. The participants were identified and recruited in four settings (in patients, OPD patients, Covid isolation facilities and the hospital waiting areas) and authors rightly explain the importance of settings where they were recruited in terms of severity and urgency of their symptoms.

Response: Thank you for the positive review of our manuscript. 

The most intriguing finding of the study is that with increasing age there was increasing trust in the doctor but less difficulties in doctor-patient communication. One would expect otherwise especially the communication barrier increasing with increasing age of the patient. The authors elaborate on three possible reasons for this finding which are interesting and plausible in the context of Indian healthcare system. However, the fact that the communication with elderly patients might always have been challenging and therefore the elderly did not experience any added barriers in communicating with the doctors raises important questions regarding doctor patient communication during non-covid times. To what extent this finding could be also influenced by the fact that the elderly in particular felt grateful to be seen by someone in a health facility in these trying times when they might have tried hard to reach the hospital for several days? Could this sense of gratitude combined with high degree of respect towards doctors in general in Indian society at least in the minds of older patients influenced their perception of reduced communication barrier? This aspect can be explored further in the discussion section.

Response: Thank you for this suggestion. Yes, it is quite possible that the gratitude that the elderly felt for having received any kind of medical attention could be a reason for them not perceiving any barriers in communication. We have included this in the discussion. 

In follow-up of the point above, healthcare providers in PPE and masks during Covid 19 pandemic probably made it impossible for the patients to assess whether the particular HCW interacting with them is a doctor, a medical student and intern or a nurse. There is hierarchy among different health care professionals and patients also respect and trust these different groups of health care professionals differently. PPE and overalls in some ways homogenized all HCWs which can otherwise be easily stratified by patients based on the age, socio-economic status based on jewelry and clothing wore under the apron etc. I wonder what impact this phenomenon (of homogenization of HCWs in PPEs) could have had on patients’ level of trust in doctors. Could authors reflect on this aspect further and particularly its implications for doctor patient relationship in non-pandemic settings.

Response: We thank the reviewer for this excellent suggestion. We agree that the PPE homogenized all the cadres of staff in a hospital. It could have had a substantial influence on the perceived barriers to communication as well as trust. We have included a sentence explaining this in the discussion. 

Women patients trusted doctors more than men. How can this be explained especially if it is not linked with education? Is it because women in general are less often in position of power or less likely to challenge the individuals with authority like doctors and therefore tend to put higher respect and trust in doctors?

It is worth noting that age, sex, occupation were not significantly associated with trust on multivariable analysis. This indicates some kind of confounding bias. Therefore we are not discussing the nuances of the gender difference in trust in detail. 

I wonder whether the authors analyzed the responses in three domains in relation to the settings in which the participants were recruited in this study. How was the experience of those seen in outpatient department different than those who were admitted or were in isolation facility or were just in waiting areas? My suspicion is that the context in which they answered the questions combined with their symptoms and severity of those symptoms might pose different challenges in doctor patient communication and might also affect the level of trust they put in their healthcare providers. I would also expect difference in challenges encountered to access health services for participants in these four distinct settings, not just in terms of reaching the healthcare facility but also once they entered the facility while they were being triaged or diverted in different streams. Could authors elaborate on why they did not pursue this line of analysis?

Response: We had originally sampled the study participants based on their location in order to perform this analysis. But to our disappointment we could not perform this analysis because the data on where they were interviewed was not captured in the dataset. Further, we did not collect data on the severity of the illness. Most of the respondents of the study were not seriously ill. We have included this in the limitation of the study. 

Reviewer #7: Dear Authors,

The submitted manuscript, with its objectives is currently of utmost importance in the COVID era. This may help in formulating policies for future pandemics.

Response: Thank you for this positive review of our manuscript. 

However, the following points need to be rectified in the manuscript

1. There are two objectives- first is to formulate a questionnaire using factor analysis and then execute the same. Hence, this needs to be clarified and subdivided in the paper

Response: We would like to clarify that the objective of the study is only to explore the communication issues and trust in doctors during the COVID 19 pandemic. Developing a questionnaire was not one of the objectives. The factor analysis was used to assign factor loadings to the various items and give them weighted scores. 

2. The flow of paper needs to be streamlined. The factor analysis to be explained first, then the execution of the study using this questionnaire.

Response: As the factor analysis was used only to assign weights to the items while scoring, it is retained in its current position. It is not done to primarily validate the questionnaire. 

3. Language and fluidity needs to be looked into.

Response: We have carefully reviewed and edited the manuscript for language and fluidity as recommended. 

Was the interview taken face to face and directly entered in google form?

Response: Yes, the interview was taken face to face and directly entered in the Google Form. 

Likert scale and scoring to be mentioned in the questionnaire part of the methodology

Response: We have now mentioned the likert and scoring in the questionnaire part of the methodology. 

Reviewer #8: The title of the study is more contemporary addressing the current issues that has struck the globe. There is an attempt with success to explore the outcomes from the COVID imposed behaviour which is also made mandatory by the Government and Statutes as 'Covid appropriate behaviour'.It is further appreciated that in a Metropolitan City of Southern India, the Covid appropriate behaviour from the part of the doctors and medical professionals had reduced the effective doctor-patient relationship. The authors could have explored the probable measures to overcome these obstacles, especially in a time when the globe is further being made to face the second wave ofthe pandemic.

Response: We thank the reviewer for the positive feedback on the manuscript. The scope of the study was limited to understanding the barriers in communication as well as trust in doctors during the COVID 19 pandemic. We were unable to explore the measures to overcome the obstacles. 

Reviewer #9: The authors have used Cronbach’s Alpha coefficient for internal consistency of parameters. This is not a statistical method but I do not know details of how this method works. I also do not know varimax rotational method although the meaning of factor analysis to study relationship of factors is understood. Application of Bartlett's test of sphericity to test whether the authors' model is fit or not is also not understood by me.

Response: Cronbach’s alpha coefficient is a statistical method to assess internal consistency reliability of a scale. It works by studying covariance patterns. Varimax rotation method is a method of rotating the factor structure such that meaningful grouping of variables can happen. Bartlett’s test of spericity is a type of Chi Square test that gives information regarding the model fit of a factor analysis model. 

I have attached my review comments. On the whole the paper appears to be fine and requires only minor corrections.

---

## [Decision Letter · Decision Letter 1]

7 Jun 2021

Doctor-patient communication and trust in doctors during COVID 19 times – a cross sectional study in Chennai, India

PONE-D-21-08356R1

Dear Dr. Gopichandran,

We’re pleased to inform you that your manuscript has been judged scientifically suitable for publication and will be formally accepted for publication once it meets all outstanding technical requirements.

Kind regards,

Prof. Ritesh G. Menezes, M.B.B.S., M.D., Diplomate N.B.

Academic Editor

PLOS ONE

Reviewers' comments:

Reviewer's Responses to Questions

**Comments to the Author**

1. If the authors have adequately addressed your comments raised in a previous round of review and you feel that this manuscript is now acceptable for publication, you may indicate that here to bypass the “Comments to the Author” section, enter your conflict of interest statement in the “Confidential to Editor” section, and submit your "Accept" recommendation.

Reviewer #6: All comments have been addressed

Reviewer #7: All comments have been addressed

Reviewer #9: All comments have been addressed

2. Is the manuscript technically sound, and do the data support the conclusions?

Reviewer #6: Yes

Reviewer #7: Yes

Reviewer #9: Yes

3. Has the statistical analysis been performed appropriately and rigorously? 

Reviewer #6: I Don't Know

Reviewer #7: Yes

Reviewer #9: (No Response)

4. Have the authors made all data underlying the findings in their manuscript fully available?

Reviewer #6: No

Reviewer #7: Yes

Reviewer #9: Yes

5. Is the manuscript presented in an intelligible fashion and written in standard English?

Reviewer #6: Yes

Reviewer #7: Yes

Reviewer #9: Yes

6. Review Comments to the Author

Reviewer #6: (No Response)

Reviewer #7: (No Response)

Reviewer #9: (No Response)

7. PLOS authors have the option to publish the peer review history of their article (what does this mean?). If published, this will include your full peer review and any attached files.

Reviewer #6: **Yes: **Priya satalkar

Reviewer #7: No

Reviewer #9: No

---

## [Editor Report · Acceptance letter]

15 Jun 2021

PONE-D-21-08356R1 

Doctor-patient communication and trust in doctors during COVID 19 times – a cross sectional study in Chennai, India. 

Dear Dr. Gopichandran:

I'm pleased to inform you that your manuscript has been deemed suitable for publication in PLOS ONE. Congratulations! Your manuscript is now with our production department. 

Kind regards, 

on behalf of

Prof. Dr. Ritesh G. Menezes 

Academic Editor

PLOS ONE